# Canine Leptospirosis in a Northwestern Region of Colombia: Serological, Molecular and Epidemiological Factors

**DOI:** 10.3390/pathogens11091040

**Published:** 2022-09-13

**Authors:** Janeth Perez-Garcia, Fernando P. Monroy, Piedad Agudelo-Florez

**Affiliations:** 1Facultad de Medicina Veterinaria y Zootecnia, Universidad CES, Medellín 050021, Colombia; 2Department of Biological Sciences, Northern Arizona University, Flagstaff, AZ 86011, USA; 3Escuela de Graduados, Universidad CES, Medellín 050021, Colombia

**Keywords:** dogs, Colombia, leptospirosis, *Leptospira santarosai*, seroreactivity, risk factors

## Abstract

Canine leptospirosis is a zoonosis of epidemiological importance. Dogs are recognized as primary reservoirs of *Leptospira interrogans* serogroup Canicola and a source of infection to the environment through urine. This study aimed to determine the presence of antibodies against *Leptospira* in canines from 49 municipalities in the Department of Antioquia, Colombia. We performed a cross-sectional study of dogs included in a neutering control program. We collected 1335 sera samples, assayed by a microagglutination test (MAT), and performed PCR detection in 21 urine samples. We also surveyed 903 dog owners. We found a seroreactivity of 11.2% (150/1335) in Antioquia with titers ≥1:50. Municipalities with the highest number of cases were Belmira (46.1%), Turbo (34.5%), and Concepción (31.0%). *L. santarosai* was identified by phylogenetic analysis in one urine sample from the municipality of Granada. The most important factor associated with a positive result was the lack of vaccination against leptospirosis (PR 3.3, *p* ≤ 0.014). Environmental factors such as water presence and bare soil around the household were also associated with *Leptospira* seroreactivity in the Department of Antioquia. We reviewed a national epidemiological surveillance database for human cases in those municipalities. We found a correlation between the high number of cases in canines and humans, especially in the Uraba. Serological and molecular results showed the circulation of *Leptospira*. Future public health efforts in the municipalities with the highest numbers of seroreactivity should be directed towards vaccination to prevent animal disease and decrease the probability of transmission of *Leptospira*. Dogs actively participate in the *Leptospira* cycle in Antioquia and encourage the implementation of vaccination protocols and coverage.

## 1. Introduction

Leptospirosis is a re-emerging zoonotic disease of global distribution caused by pathogenic spirochetes of the genus *Leptospira*, which contains more than 300 pathogenic serovars. Pathogenic species can remain in the water and alkaline soils but are mostly confined to the kidneys of a wide range of hosts. Humans, some domestic animals, and wildlife are highly susceptible to *Leptospira,* presenting mild and moderate clinical manifestations that may have a fatal outcome. Other animals have adapted to some serovars or serogroups of *Leptospira* and develop clinical manifestations that tend to be asymptomatic and chronic. They are considered maintenance hosts and disseminators of the bacterium and can shed on occasion throughout their lifetime. Urine in these infected animals is the primary source of infection for susceptible animals [1,2].

Canine leptospirosis is widespread worldwide, and dogs are incidental hosts for some *Leptospira* serovars and maintenance hosts for *L. interrogans* serovar Canicola. The relationship between leptospirosis in humans and their dogs could provide evidence of intra- and interspecific transmission or exposure to the same risk factors [3,4].

Dogs with leptospirosis can manifest clinical jaundice, uremia, or acute hemorrhagic diathesis [5]. The Canicola serogroup causes most clinical cases in canines, first reported in 1933 with the first isolate of the strain Hond Ultrech IV [2]. Dogs infected with this serogroup present a mild to moderate symptomatology, and they recover without relevant sequelae. Other *Leptospira* serogroups, mainly Icterohaemorrhagiae, cause severe and fatal canine leptospirosis cases. Canine leptospirosis has been considered a re-emergent disease in some countries due to changes in the serogroups responsible, clinical signs, and disease outcomes [5,6]. The serogroups Canicola and Icterohaemorrhagiae are the most commonly found in dogs and are used widely in vaccines. Vaccination has decreased the worldwide incidence of canine leptospirosis by these serovars in the last two decades. Vaccination prevents canine disease and subsequent zoonosis transmission to humans. However, changes in the incidence of serogroups affecting dogs have resulted in low effectiveness for canine vaccination when new serovars have been included in commercially available vaccines [7,8]. Additionally, vaccines would be more effective with knowledge of the serogroups of *Leptospira* circulating in the geographical area where the canine population originated [9].

In Colombia, the pathogenic group *Leptospira interrogans* sensu lato is the most frequent serogroup in canine leptospirosis recorded since 1966. These serogroups include Canicola, Ballum, Pyrogenes, and Icterohaemorrhagiae, and in the last few years, Pomona, Grippotyphosa, Australis, Sejroe, and Panama [10,11,12,13,14,15]. From the perspective of veterinary medicine and public health, it is necessary to identify the serogroups circulating in local areas and recognize the role of dogs in the epidemiology of the disease. This study aimed to determine the presence of the two main serogroups of *Leptospira* in dogs (Canicola and Icterohaemorrhagiae) in asymptomatic canines from 49 municipalities in the Department of Antioquia, Colombia. Using serology and molecular testing, we will explore other serogroups circulating in this region, describe epidemiological risk factors through owners self-reporting characteristics, and correlate with human leptospirosis cases reported in the National Surveillance System in Antioquia.

## 2. Results

Canines were 78.3% female, and although canines of 23 different breeds were included, the highest proportion was mixed breeds, at 73.3%. Dogs from urban areas comprised 57.3%.

Seroreactivity was 11.2% (150/1335), with positive MAT to at least one of the evaluated serogroups. Figure 1 shows the frequency of positive reactive cases by sub-region in the Department of Antioquia.

The highest percentage was obtained in the Uraba region (22.4%) with 35 seropositive dogs, and the lowest in the west region with 17 dogs (7.3%). Of the 49 municipalities, 24 presented seropositivity above the level obtained for the entire Department (11.2%), as shown in Table 1.

The municipality with the highest seroreactivity was Belmira (46.15%), followed by Turbo (34.4%) and Concepción (31.0%). Ten municipalities did not have any canines with a positive microagglutination test, as shown in Table 1. We found a statistically significant association between the presence of antibodies against *Leptospira* and municipalities (*p* ≤ 0.000) as well as the study regions (*p* ≤ 0.000).

The proportion of the sampled animals included in the study was calculated with the reported canine population by municipality (Figure 2). We found no statistical correlation between the presence of reactive canines in the municipalities and the incidence of human cases (*p* ≤ 0.1563). However, a significant relationship existed between municipalities of the Urabá region, with more than 19 cases per 100,000 inhabitants and almost 40% of canine reactive cases.

### 2.1. Serogroups Circulating

We analyzed all samples (1335) against the Canicola and Icterohaemorrhagiae serogroups. Serogroup Canicola had the highest number of cases, 50.6% (76/150), and the highest antibody titers, 1:3200. Furthermore, 28.6% (43/150) of seropositive dogs were reactive to the Icterohaemorrhagiae serogroup, with titers between 1:100 and 1:200 (Figure 3).

A set of 212 randomly selected sera from all regions was processed with a panel of nine serogroups, including a local strain JET, as described in Table 2. Extended MAT results include *L. interrogans* serovar Pomona (8/150), *L. interrogans* serovar Hardjo (5/150), *L. kirschneri* serovar Grippothyposa (3/150), *L. borgpetersenii* serovar Tarassovi (2/150), *L. borgpetersenii* serovar Ballum (6/150), *L. borgpetersenii* serovar Bratislava (3/150), and *L. santarosai* serovar Alice strain JET (4/150). Coaglutination defined as positive to both serogroups was found in 30 dogs.

Including all the tested serogroups, the percentage of seroreactivity was increased by 0.74% for ten additional positive canines (150/1335). We found two samples with co-agglutination between *L. Interrogans* serogroup Canicola (strain Utrecht) and *L. santarosai* serogroup Grippothyposa (strain JET).

### 2.2. Molecular Characterization

Of 21 urine samples collected, only one was positive by PCR for *Leptospira* 16S gene. This male mix breed dog lived in an urban household in the municipality of Granada, subregion Eastern region; it was 30 months old, seronegative, and had no history of the disease or vaccination. The dog spent most of the time outside the home and with other canines. The molecular analysis allowed us to define a 99% similarity to *Leptospira santarosai*.

### 2.3. Owners, Housing, and Dog Information

Only in two dogs did the owners report a history compatible with leptospirosis. Both cases were negative in the microagglutination test. The median age for reactive dogs was 24 months and 18 months for the non-reactive dogs, which was found to be a statistically significant difference (*p* ≤ 0.029).

When looking at signs of disease as presented by owners, none of the animals presented orchitis, three reported hematuria (0.39%; 3/770), five reported jaundice (0.67%, 5/742), and 15 had abortions (2.48%; 15/603). Reports of abortion presented three times more in seroreactive dogs than in those who had a negative MAT test (CI 95% 1.27–5.66, *p* ≤ 0.034).

Vaccination against leptospirosis was reported by 67 owners, equivalent to 8.47%, as shown in Table 3. Non-vaccinated MAT reactive dogs comprised 14.09% (102/724), and MAT reactive and vaccinated comprised 4.48% (3/67). Association between positivity and vaccination status was statistically significant (*p* ≤ 0.014).

We evaluated the interaction of dogs with their owners through time spent inside households; 14.71% (64/435) of dogs that remained inside the house most of the day were MAT positive, and 14.94% (49/328) slept inside the owner’s room.

When evaluating the presence of animals in the same household, 86% of the dogs shared with other potential *Leptospira* animal reservoirs: 92% cohabited with other canines and 51.3% with felines. Of the seroreactive dogs, 29.2% lived in homes with self-reported presence of rodents. Figure 4 describes the percentages of coexistence with all animal species as reported by owners. We also included the number of reactive canines by the microagglutination test. It should be noted that the largest number was recorded for the serogroup Icterohaemorrhagiae for intra-species coexistence and the Canicola serogroup for both intra- and inter-species coexistence.

We found that 65.4% of the canine owners were females. The most frequent occupations reported were “housewife” and “retired”, at 45.3%, and 12.8% were workers in the agricultural or primary economic sectors. Only one of the owners reported clinical symptoms compatible with leptospirosis. She lived in the urban area of the municipality of Turbo. Her pet was a 3-year-old female with no clinical history of the disease and positive microagglutination for the Canicola and Tarassovi serogroups, with titers of 1:100 and 1:200, respectively.

Housing characteristics of MAT reactive dogs include 14.06% with concrete or tile roofing, 15.15% with concrete walls, and 14.25% with ceramic and concrete flooring. However, dirt or wood flooring (traditional materials) were more common in reactive dogs than in those with negative MAT (*p* < 0.06). The peridomiciliary environment for the reactive dogs was mainly water sources (30.61%), forests (23.53%), and crops (16.42%). Water sources were 2.2 times more frequent in seroreactive dogs (CI 95% 1.29–3.62; *p* < 0.006). Other peridomiciliary characteristics in MAT reactive dogs were the presence of roads in 23.60% of dwellings and bare soil in 34.78% (Table 4). Seroreactive dogs were three times more frequent in households with an outdoor kitchen than in houses where food preparation was done in an indoor kitchen as an independent house space (*p* < 0052). Although most houses had municipal services for garbage collection, aqueducts, and sewerage, there were more cases of seroreactive dogs in homes with open field garbage and human waste disposal.

## 3. Discussion

The presence of canine leptospirosis in Colombia is known and commonly associated with the Canicola serogroup. In the Department of Antioquia, this study reports the presence of antibodies in 11.2% of asymptomatic canines by MAT. In addition, we found evidence of other circulating serogroups in the pathogenic group *Leptospira interrogans* serogroups Icterohaemorrhagiae, Pomona, Sejroe, Grippothyposa, Tarassovi, Ballum, and Australis. The presence of *Leptospira* antibodies in dogs highlights the circulation of this microorganism in susceptible hosts. Dogs could be indicators of *Leptospira* intra- or inter-specific infection.

Our results provided molecular evidence for the presence of *Leptospira santarosai* in the urine of an asymptomatic dog. There is only one known report on this *Leptospira* species in dogs in Colombia [16]. This finding suggests that canine leptospirosis in Antioquia has a unique and changing dynamic from a classic to a re-emergent presentation, as evident in other Latin American regions such as Brazil [17].

We found variable seropositivity of canine leptospirosis by MAT in 49 municipalities of the Department of Antioquia when compared to previous reports in other cities and regions in Colombia. It was lower than the highest reported in an indigenous territory in Colombia of 79.9% [18] but higher than the last report for the principal city of the same Department of this study of 8.4% [19]. A limitation in our results is that analysis must be done by region or municipality. This is an important factor because there may be considerable differences in the vaccination status and epidemiological characteristics. In addition, we must consider differences in climate, environmental and geographical conditions, and the diversity of domestic, livestock, and wild animal species.

A concrete example is the Uraba region, which had seropositivity of 22.4%. This highlights the highest frequencies per municipality, such as Turbo (proportion of sampled canines by municipality of 0.76%) with 34.4%, with 14 positive results of 31 sampled canines. Additionally, Mutata, Chigorodo, Carepa, San Pedro de Uraba, and Arboletes had frequencies between 20 and 30% in the same area. In 2007, a study on the human population reported a seroprevalence of 12.5% and positivity with serogroups associated with rodents and canines [20]. Previous studies in this area had generated consideration of the ownership of pets and wild animals [21,22] as a risk factor for disease in humans due to the isolation of *L. santarosai* in humans and canines [16]. However, there were no previous reports of seroprevalence of *Leptospira* in canines in this region. An outbreak was reported by the Colombian military forces in this area, describing a zoonotic link between six affected people who had contact with a canine presenting clinical signs and who were later diagnosed with the disease.

The serological and molecular results obtained in the present study provided evidence of the circulation of this organism in this susceptible population, and vaccination would have a high probability of preventing and decreasing the transmission of *Leptospira* in Antioquia. An important finding from the geographical analysis pertains to the municipality of Belmira, in the northern region of the Department. It had the highest canine seroprevalence of all the municipalities included in the study (46.15% of a proportion sampled for this municipality of 1.39%). This municipality is known for having many bovine farms, with the recognized seroprevalence in this species. However, according to official reports from the National Surveillance System in 2015, this municipality did not report human leptospirosis cases between 2007 and 2015. It is essential to actively search for cases in different animal populations and humans.

We found an absence of human cases in the Northeast municipalities included in the study, with a prevalence of 8.7% against zero cases of human leptospirosis in 2015. In this study, six of the ten municipalities that did not report canines with positive microagglutination tests did not report human leptospirosis cases in 2015 in the National Surveillance System. This absence could be due to a lack of circulation of *Leptospira* in these municipalities or case underreporting, since they meet all the environmental, demographic, and epidemiological risk factors for leptospirosis transmission. The proportion of canines sampled in some municipalities (Table 2) could represent an information bias that does not allow us to draw clear-cut conclusions on the actual status of the municipalities in potential epidemiological underreporting.

The Canicola serogroup was the most common serogroup found, as we expected, in dogs [2]. Furthermore, the presence of the Icterohaemorrhagiae serogroup confirms what is evident in other regions of Colombia: the epidemiological nexus of cases with typical synanthropic sources of infection such as rodents [10,11,12,13,14,15]. Titers might indicate that these dogs were susceptible hosts and may be asymptomatic while affected by the bacterium. Dogs can have greater exposure to this rodent-related serogroup than humans because they have free access to urine-contaminated environments, hunt rodents, and often their drinking water can be contaminated with the urine of these synanthropic species. A study in Brazil found evidence of the relationship between serovars and MAT titers, concluding that the Canicola serogroup could have higher titers than the Icterohaemorrhagiae serogroup [23,24].

The present study illustrates the great variety of serovars found in studies of the prevalence of *Leptospira* in a highly biodiverse country such as Colombia. Finding a pathogenic *Leptospira santarosai* by molecular techniques in an asymptomatic canine should be further studied. MAT results need to be correlated with direct techniques, because there are extrinsic and intrinsic factors that can produce variability in titers [25].

Vaccination against leptospirosis was reported by 67 owners, equivalent to 8.47% (67/791). Our MAT test found that 97.10% of positive dogs were not vaccinated (102/105), while only 4.48% of the vaccinated animals had antibodies in the microagglutination test (3/67). We found a low rate of seroreactivity in vaccinated dogs compared with those that were unvaccinated. This association was statistically significant (*p* < 0.029). In this study, titers in vaccinated dogs did not exceed 1:200, a value not associated with recent vaccination [26].

The frequency of *Leptospira* by PCR in urine samples was considerably low. Although low, this finding suggests the latent risk that humans become infected by direct or indirect contact with the urine of infected animals. Most canines (86%) coexisted with at least one other animal in the same house—mainly dogs, cats, and rodents. The proximity between canines and rodents is a risk factor for disease transmission in urban environments. A study in Brazil reported the behavior of “hunting mice” as an important risk factor in canine leptospirosis [27]. Therefore, it is important to continue an integrated control of rats and mice and the follow-up of reservoirs that have historically been considered the primary source of *Leptospira* infection in humans and other animals.

Dogs’ natural and instinctive behaviors could be risk factors, including urine marking, drinking water outdoors from an untreated source, and hunting small mammals (rodents, bats, or marsupials). We explored the presence of natural and untreated water sources in the peridomicile as a risk factor for canine leptospirosis (RP 2.46; *p* < 0.038) as well as the risk factors of being exposed to high urine contamination and risk interactions such as playing, swimming, and drinking. As an indirect measurement of the interaction between dogs and their owners, we identified the percentage of canines sharing the same space at night with their owners. It was high in this study (45.3%), promoting contact situations with urine, such as marking territory, using pee pads, sniffing, licking genitals, or locating a humid area within the dwelling. Those practices facilitate the microorganism spreading quickly [28]. Moreover, the probability of infection could be higher, not only for humans but also for other animals in the same household.

Canines should be considered sentinel species of leptospirosis in the human population. Active surveillance could help in the early detection of sources of *Leptospira*, such as infected animals and contaminated water. In addition, direct interventions such as canine vaccination, antimicrobial treatment in susceptible species, and implementation of control measures for reservoirs such as rodents are likely to minimize the risk factors that increase disease presentation in susceptible species.

## 4. Materials and Methods

### 4.1. Ethical Considerations

The study was approved by the Institutional Committee for the Care and Use of animals (CICUA) of Universidad CES, Acta 16 of 20 October 2015. All owners signed informed consent before collecting the sample and conducting surveys.

### 4.2. Study Area

We included surveys of 49 municipalities distributed in seven regions of the Department of Antioquia. The study was developed during a neutering program, “Animóvil”, sponsored by the Antioquia Government.

### 4.3. Type of Study

A cross-sectional study was carried out in which we obtained 1335 blood and 21 urine samples. We also administered 903 surveys to dog owners who voluntarily participated in the animal neutering program or were responsible for two of the municipal shelters included in the study performed in 2015. Some owners were responsible for more than one animal.

### 4.4. Procedures

A group of veterinarians evaluated all dogs to identify clinical signs or abnormalities. Only healthy canines were included in the study. We included all dogs whose owners allowed collection of samples, and urine samples were taken from animals that had urine in the bladder or spontaneously urinated at the clinical examination. Blood samples were taken from the cephalic vein (ADC BD Yellow Cap Tube, BD Vacutainer^®^, Fisher Scientific, Pittsburg, PA, USA), a urine sample was collected, and 1–2 mL of each was placed in a sterile vial. Both samples were kept for a maximum of four hours at room temperature until the completion of the entire sampling for each day. They were later stored at 4 °C until processing at the laboratory. A survey was also performed to obtain dog information and leptospirosis vaccination status and explore household risk factors.

### 4.5. Microagglutination Test (MAT)

All MATs were performed when the final collection of serum samples was completed. Samples were stored at −20 °C until processed and thawed in a cold bath for at least 15 min before performing the test. We tested all samples against the conventional canine serogroups, Canicola and Icterohaemorrhagiae. Each test included a positive control serum (internal control of the test for each serogroups) and negative control (only phosphate buffer saline pH 7.4, PBS). We performed serial dilutions with PBS (1:25) to determine antibody titers, beginning with 1:25. A 1:50 dilution was used as a cutoff for seroreactivity [28]. Tests were read after one-hour incubation using a dark field microscope with a 4× objective without a coverslip; agglutination of 50% of the field was used as a positivity indicator in a sample of 20 μL of the antigen-serum mixture.

Additionally, we tested a random selection of 212 sera samples (15% of total samples) against a panel of eight reference serogroups (Pomona, Sejroe, Grippothyposa, Tarassovi, Ballum, and Australis). We also included a local *Leptospira* strain belonging to the Grippothyphosa serogroup of *Leptospira santarosai*. A full description of the panel used for evaluation is presented in Table 2.

### 4.6. Molecular Characterization

We performed DNA extraction of urine samples using a commercial kit (Wizard Genomic DNA purification Kit^®^. Promega Corp., Madison, WI. USA). Samples were stored at 4 °C in Tris-EDTA buffer until processing. PCR was used to amplify the 16S ribosomal gene, and products were separated by electrophoresis in agarose gels as previously reported [15]. Products were subsequently sequenced (Macrogen Inc^®^. Seoul, Korea 2016) to determine the *Leptospira* sp.

### 4.7. Information Analysis

Surveys were analyzed using the IBM^®^ SPSS^®^ Statistics 21.0 (IBM Corporation^®^. Armond, NY, USA. Universidad CES license) and Epidat^®^ 3.1 and 4.2 (Organización Panamericana de la Salud (OPS-OMS), Universidad CES. Coruña, España 2016). We performed a descriptive analysis by municipality according to the case definition: seroreactivity for the serogroups Canicola and Icterohaemorrhagiae with titers over 1:50. A Chi-square test or Fisher test association analysis analyzed the significance between the MAT positivity and epidemiological variables. Quantitative variables were tested using the Shapiro-France test and Mann-Whitney U test to determine correlation with the presence of disease.

The Antioquia Government provided animal populations. In each municipality, epidemiological surveillance information of human cases of leptospirosis included in the study was downloaded from the official website to correlate with the proportion of positive animals per place. The Spearman correlation test analyzed the significance between both variables. All tests used a 95% significance level, and statistical significance was assessed with a *p* < 0.05 value. We used free Piktochart (Piktochart Sdn. Bhd. Penang, Malaysia. www.piktochart.com, accessed on 11 September 2022) to create graphs and maps designed in ArcGIS^®^ 10.4 (Esri Redlands, CA, USA. Universidad CES license).

## Figures and Tables

**Figure 1 pathogens-11-01040-f001:**
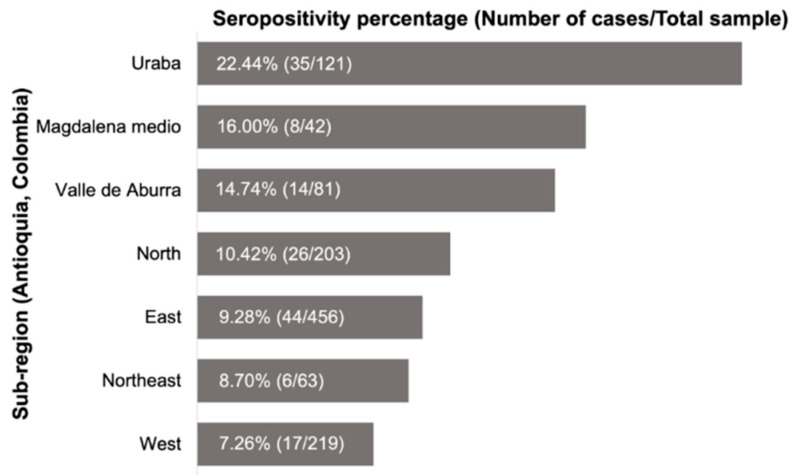
Distribution percentage of canine leptospirosis obtained by subregion in the Department of Antioquia. The right side shows the number of reactive canines by microagglutination and the total number of dogs included in the study by region (Antioquia, 2015).

**Figure 2 pathogens-11-01040-f002:**
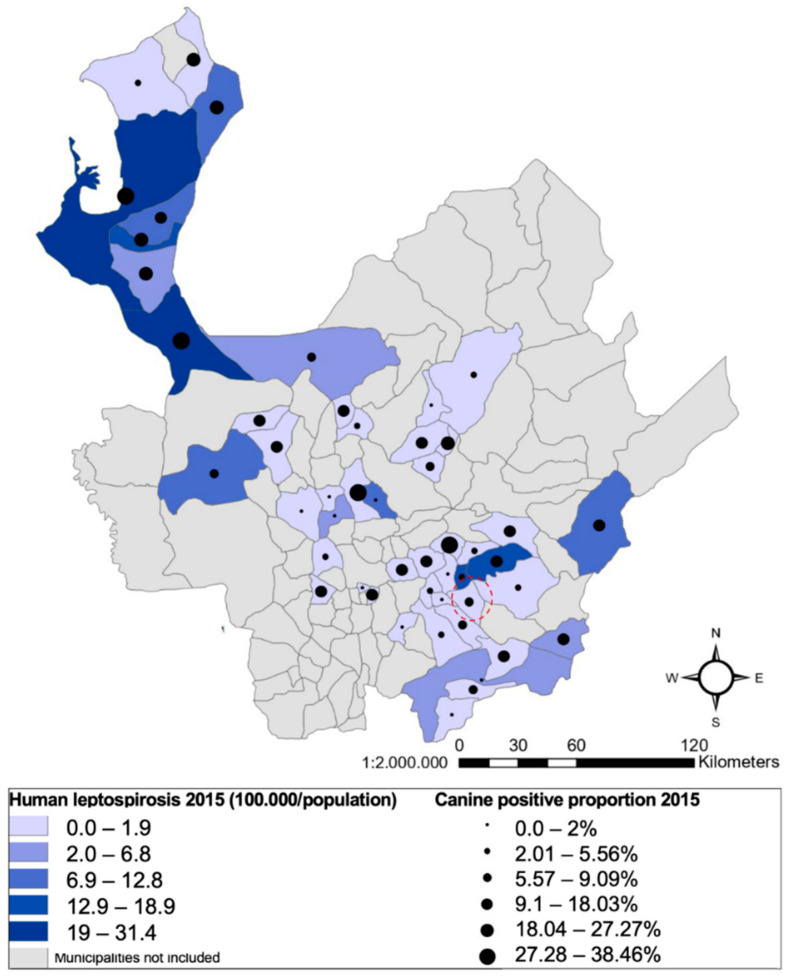
The geographical distribution of the percentages of seropositivity to canine leptospirosis and human leptospirosis reported in the Department of Antioquia in 2015 by the National Surveillance System. The blue color shows human case distribution (rates per 100,000 inhabitants). Points denote canine cases obtained in the present study.

**Figure 3 pathogens-11-01040-f003:**
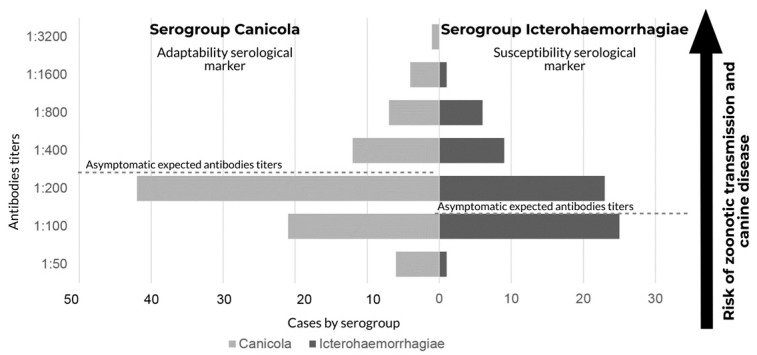
Distribution of seroreactive cases based on antibody titers in the microagglutination test (MAT). Reactivity against serogroup (Canicola) was used as an adaptability marker of reservoirs and the susceptibility serogroup (Icterohaemorrhagiae) as an incidental host in Antioquia 2015. The dotted line describes hypothetical antibody titers for each serogroup in asymptomatic canines. The presence of high antibody titers suggests canine leptospirosis.

**Figure 4 pathogens-11-01040-f004:**
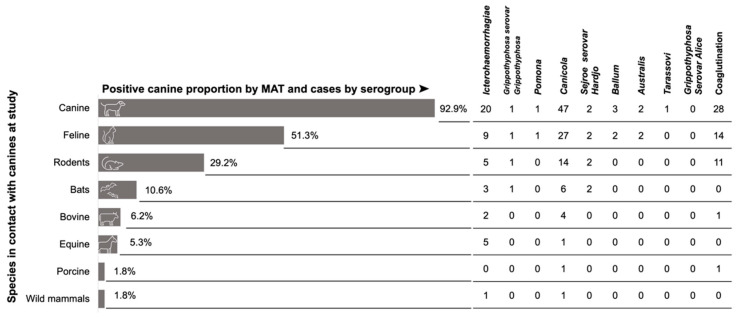
The proportion of seroreactive dogs cohabiting in the same household with other animals and showing positivity in the microagglutination test. The value in each row indicates the number of dog positives for the serogroup. Co-agglutination was included when there was reactivity to more than two serogroups at the same titer and was evaluated separately.

**Table 1 pathogens-11-01040-t001:** Description of the microagglutination test results found in each of the municipalities. Samples are described by municipality with respect to the data of the canine census and the proportion sampled by the municipality in Antioquia, 2015.

Municipality	Reactive MAT		Not Reactive MAT		Sample by Municipality	Canine Population by Municipality	Proportion Sampled by Municipality
	n	%	n	%	n	n	%
**Uraba region**							
Apartadó	2	11.76%	15	88.24%	17	3369	0.50%
Arboletes	3	20.00%	12	80.00%	15	2930	0.51%
Carepa	3	27.27%	8	72.73%	11	1642	0.67%
Chigorodó	4	25.00%	12	75.00%	16	4026	0.40%
Mutatá	8	29.63%	19	70.37%	27	1280	2.11%
Necoclí	2	6.67%	28	93.33%	30	3680	0.82%
San Pedro de Urabá	2	25.00%	6	75.00%	8	2400	0.33%
Turbo	11	34.38%	21	65.63%	32	4220	0.76%
**Magdalena medio region**							
Puerto Berrio	3	16.67%	15	83.33%	18	2375	0.76%
Puerto Triunfo	5	15.63%	27	84.38%	32	1120	2.86%
**Valle de Aburra region**							
Envigado	14	17.50%	66	82.50%	80	7350	1.09%
Itagüí	0	0.00%	15	100.0%	15	8690	0.17%
**North region**							
Angostura	3	11.54%	23	88.46%	26	1127	2.31%
Belmira	6	46.15%	7	53.85%	13	933	1.39%
Campamento	0	0.00%	29	100.0%	29	1130	2.57%
Carolina	3	9.09%	30	90.91%	33	2020	1.63%
Entrerrios	0	0.00%	21	100.0%	21	1024	2.05%
Guadalupe	4	20.00%	16	80.00%	20	594	3.37%
Ituango	1	6.25%	15	93.75%	16	1450	1.10%
San Andrés	1	5.56%	17	94.44%	18	788	2.28%
Toledo	2	12.50%	14	87.50%	16	375	4.27%
**Eastern region**							
Alejandría	1	2.86%	34	97.14%	35	594	5.89%
Argelia	2	8.33%	22	91.67%	24	1010	2.38%
Cocorná	4	12.50%	28	87.50%	32	936	3.42%
Concepción	9	31.03%	20	68.97%	29	1100	2.64%
El Carmen	2	5.41%	35	94.59%	37	2800	1.32%
El Peñol	1	2.00%	49	98.00%	50	2669	1.87%
Granada	2	8.33%	22	91.67%	24	600	4.00%
Guarne	2	13.33%	13	86.67%	15	2815	0.53%
Guatapé	1	3.57%	27	96.43%	28	843	3.32%
La Ceja	0	0.00%	19	100.0%	19	2100	0.90%
Marinilla	1	4.17%	23	95.83%	24	2825	0.85%
Nariño	0	0.00%	23	100.0%	23	1250	1.84%
Santuario	0	0.00%	18	100.0%	18	1800	1.00%
San Carlos	2	5.56%	34	94.44%	36	1240	2.90%
San Francisco	3	14.29%	18	85.71%	21	463	4.54%
San Rafael	14	22.95%	47	77.05%	61	1410	4.33%
San Vicente	6	16.22%	31	83.78%	37	1929	1.92%
Sonsón	0	0.00%	26	100.0%	26	4357	0.60%
**Northeast region**							
Anorí	1	3.03%	32	96.97%	33	801	4.12%
San Roque	5	13.89%	31	86.11%	36	1920	1.88%
**West region**							
Armenia	7	17.50%	33	82.50%	40	604	6.62%
Cañas Gordas	3	14.29%	18	85.71%	21	1398	1.50%
Ebéjico	1	4.35%	22	95.65%	23	2450	0.94%
Frontino	3	7.32%	38	92.68%	41	1593	2.57%
Olaya	0	0.00%	22	100.0%	22	570	3.86%
Santafé de Antioquia	0	0.00%	27	100.0%	27	2050	1.32%
Sopetran	0	0.00%	48	100.0%	48	1470	3.27%
Uramita	3	25.00%	9	75.00%	12	890	1.35%
**Total, municipalities**	**150**	**11.24%**	**1185**	**88.76%**	**1335**	**97,010**	**1.38%**
**under study**							

**Table 2 pathogens-11-01040-t002:** Description of the panel of *Leptospira* sp. serogroups used in the microagglutination test (MAT) and the number of positive dogs with titers ≥ 1:50 in the Department of Antioquia. Canicola and Icterohaemorrhagiae serogroups were tested in 1335 dogs and other serogroups in a subset of 212 dogs.

No.	Species	Serogroup	Serovar	Strain	Number of Positives
1	*L. interrogans*	Icterohaemorrhagiae	Icterohaemorrhagiae	RGA	43
2	*L. interrogans*	Canicola	Canicola	Hond Utrecht IV	76
3	*L. interrogans*	Pomona	Pomona	Pomona	8
4	*L. interrogans*	Sejroe	Hardjo	Hardjoprajitno	5
5	*L. interrogans*	Grippothyposa	Grippothyposa	Moskva	3
6	*L. borgpetersenii*	Tarassovi	Tarassovi	Perepelitsin	2
7	*L. borgpetersenii*	Ballum	Castellonis	Castellon 3	6
8	*L. borgpetersenii*	Australis	Bratislava	Jez Bratislava	3
9	*L. santarosai*	Grippothyposa	Alice	JET	4

**Table 3 pathogens-11-01040-t003:** Prevalence ratio (PR) of the dog characteristics by the MAT results with the serogroups Canicola and Icterohaemorrhagiae, Antioquia, 2015.

	Reactive MAT		Not Reactive MAT		Total †		PR (CI 95%)	*p*-Value
	n	%	n	%	n	%		
**Sex**								
Male	25	13.44%	161	86.56%	186	100%	1.075 (0.79–1.63)	0.4080
Female	84	12.50%	588	87.50%	672	100%		
**Breed**								
Purebred	25	11.52%	192	88.48%	217	100%	0.85 (0.558–1.293)	0.2610
Cross	81	13.57%	516	86.43%	597	100%		
**Housing area**								
Rural	44	13.54%	281	86.46%	325	100%	1.014 (0.959–1.072)	0.3540
Urban	54	12.36%	383	87.64%	437	100%		
**History of leptospirosis**								
Yes	0	0.00%	2	100.00%	2	100%		0.7490
No	102	13.51%	653	86.49%	755	100%		
**Vaccination against leptospirosis**								
Yes	3	4.48%	64	95.52%	67	100%	0.318 (0.104–0.975)	**0.014** *
No	102	14.09%	622	85.91%	724	100%		
**History of clinical signs**								
**Jaundice**								
Yes	1	20.00%	4	80.00%	5	100%	1.504 (0.258–8.766)	0.5120
No	98	13.30%	639	86.70%	737	100%		
**Hematuria**								
Yes	0	0.00%	3	100.00%	3	100%		0.6530
No	102	13.30%	655	85.40%	767	100%		
**Abortion**								
Yes	5	33.33%	10	66.67%	15	100%	2.685 (1.272–5.668)	**0.034** *
No	73	12.41%	515	87.59%	588	100%		
**Canine habits**								
**Most frequented place**								
Interior	64	14.71%	371	85.29%	435	100%	1.37 (0.908–2.067)	0.0800
Exterior	29	10.74%	241	89.26%	270	100%		
**Resting place**								
Exterior	43	11.50%	331	88.50%	374	100%	0.77 (0.525–1.127)	0.1080
Interior	49	14.94%	279	85.06%	328	100%		

† There are missing values in surveys. Absolute number of total patients can change between each variable. * Value *p* < 0.05.

**Table 4 pathogens-11-01040-t004:** Prevalence ratio (PR) of the peri- and domiciliary characteristics of dog owners’ households and seropositive dogs by microagglutination test with the serogroups Canicola and Icterohaemorrhagiae in Antioquia, 2015.

	Reactive MAT		Not Reactive MAT		Total †		PR (CI 95%)	*p*-Value
	n	%	n	%	n	%		
**Roof materials**								
Modern materials	44	14.06%	269	85.94%	313	100%	0.714 (0.408–1.48)	0.164
Traditional materials	13	19.70%	53	80.30%	66	100%		
**Wall materials**								
Modern materials	45	15.15%	252	84.85%	297	100%	1.010 (0.562–1.817)	0.566
Traditional materials	12	15.00%	68	85.00%	80	100%		
**Floor materials**								
Modern materials	50	14.25%	301	85.75%	351	100%	0.509 (0.258–1.003)	0.066
Traditional materials	7	28.00%	18	72.00%	25	100%		
**History of flooding**								
Yes	5	27.78%	13	72.22%	18	100%	1.670 (0.772–3.641)	0.176
No	78	16.63%	391	83.37%	469	100%		
**Peridomiciliary housing**								
Yes	40	15.04%	226	84.96%	266	100%	0.602 (0.347–1.043)	0.064
No	13	25.00%	39	75.00%	52	100%		
**Peridomiciliary forest**								
Yes	12	23.53%	39	76.47%	51	100%	1.532 (0.867–2.708)	0.112
No	41	15.36%	226	84.64%	267	100%		
**Peridomiciliary crops**								
Yes	11	16.42%	56	83.58%	67	100%	0.981 (0.535–1.800)	0.558
No	42	16.73%	209	83.27%	251	100%		
**Peridomiciliary water**								
Yes	15	30.61%	34	69.39%	49	100%	2.167 (1.296–3.624)	**0.006** *
No	38	14.13%	231	85.87%	269	100%		
**Peridomiciliary Bare soil**								
Yes	16	34.78%	30	65.22%	46	100%	2.557 (1.558–4.200)	**0.001** *
No	37	13.60%	235	86.40%	272	100%		
**Peridomiciliary roads**								
Yes	21	23.60%	68	76.40%	89	100%	1.689 (1.031–2.764)	**0.031** *
No	32	13.97%	197	86.03%	229	100%		
**Source of drinking water**								
Treated water	71	15.71%	381	84.29%	452	100%	1.055 (0.673–1.652)	0.467
Untreated water	21	14.89%	120	85.11%	141	100%		
**Place of food preparation**								
Outside of the house	86	15.28%	477	84.72%	563	100%	0.306 (0.134–0.696)	0.052
Inside the house	3	50.00%	3	50.00%	6	100%		
**Municipal garbage collection service**								
Yes	72	15.55%	391	84.45%	463	100%	0.855 (0.569–1.285)	0.266
No	26	18.18%	117	81.82%	143	100%		
**Municipal sewer service**								
Yes	61	15.06%	344	84.94%	405	100%	0.846 (0.577–1.240)	0.231
No	34	17.80%	157	82.20%	191	100%		

† There are missing values in surveys. Absolute number of total patients can change between each variable. * Value *p* < 0.05.

## Data Availability

Nucleotide sequence from the single *Lepstopira* 16S positive PCR may be made available upon request.

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
