# Peer review of "Canine Leptospirosis in a Northwestern Region of Colombia: Serological, Molecular and Epidemiological Factors"

_pathogens, 2022, doi:10.3390/pathogens11091040_

Round 1
Reviewer 1 Report
Dear editor, authors,
this paper describes a study conducted on more than 1000 healthy dogs in Antioquia (Colombia), using a serological approach on sera and a molecular test on urines. It reports the geographical distribution of seropositivity and the authors elaborate the data obtained in order to evaluate the correlation between the positivity and various factors (vaccination status, sex, breed, frequented place, peridomiciliary environment, etc..). I think that the experimental design is appropriate but the data obtained were not interpreted appropriately (more details in major revisions) and so some statements/conclusions were not supported by the results.
I recommend reviewing the data analysis and rewriting the paragraph 2.3 and discussion sections.
For this reason I recommend a new submission of the article.
Major
Figure 2: It would be interesting to add the names of the municipalities or at least of the sub-regions. In this way the reader can better interpret the map.
Line 107-110: The percentages do not match the numbers. 93/150=62% and not 69.6%, 48/150=32% (73%?). The total number of positives sera is 149 (93+48+8) and one dog is missing.
Line 113-116: It should be moved in the results (and not in the capture) and “the asymptomatic expected antibody titers” need to be explained better, even with a citation (if there is one).
Table 3, Table 4, Paragraph 2.3: the percentages must be normalized to the relative abundance of the sampled animals. For example: the percentage of positive males is 13% (25/186) and females 12.5% (84/672) and not 22.90% and 77.10%, respectively, as instead the authors have elaborated. This is valid for all the considered characteristics in table 3 and table 4.
If reactive animals were considered to be the 100%, as done by the authors, the evaluation is compromised and untrue because it is distorted by the relative abundance of each characteristic.
All statements about this data must be corrected, as well as subsequent conclusions.
Minor
Line 33: I would erase “spp.”
Line 75-77: I would put it in materials and methods
Figure 1: The numbers indicated in brackets refer to pos/neg and not to pos/total as stated.
Line 121: “The percent” should be replaced by “the percentage”
Author Response
REVIEWER 1
Major:
-Figure 2: It would be interesting to add the names of the municipalities or at least of the sub-regions. In this way the reader can better interpret the map.
While this is a great comment, we can not add the names of the municipalities or the sub-regions because it resulted in a very crowded and busy figure. That was why, we left as it is. We are sorry about this positive suggestion, but we could noy implemented it.
-Line 107-110: The percentages do not match the numbers. 93/150=62% and not 69.6%, 48/150=32% (73%?). The total number of positives will be 149 (93+48+8) and one dog is missing.
The percentages were corrected and the results with the dominant serovar are detailed. The results of the other serovars are included in the extended version of the MAT. Coagglutination was reported for all microagglutination tests.
-Line 113-116: It should be moved in the results (and not in the capture) and “the asymptomatic expected antibody titers” need to be explained better, even with a citation (if there is one).
We changed the term used in the figure title. Also, we added a reference and a component in the discussion.
-Table 3, Table 4, Paragraph 2.3: the percentages must be normalized to the relative abundance of the sampled animals. For example: the percentage of positive males is 13% (25/186) and females 12.5% (84/672) and not 22.90% and 77.10%, respectively, as instead the authors have elaborated. This is valid for all the considered characteristics in table 3 and table 4.
The change was made.
-If reactive animals were considered to be 100%, as done by the authors, the evaluation is compromised and untrue because it is distorted by the relative abundance of each characteristic. All statements about this data must be corrected, as well as subsequent conclusions.
Maybe we don't understand this suggestion. This is because in the surveys there are missing values for each of the characteristics. This would be solved with the correction of the percentages by rows. Similarly, we added an explanatory note as a table footer.
Minor
-Line 33: I would erase “spp.”
Done
-Line 75-77: I would put it in materials and methods
It was placed here because this calculation was the one used specifically for Figure 2.
-Figure 1: The numbers indicated in brackets refer to pos/neg and not to pos/total as stated.
Is the total of animals by region
-Line 121: “The percentage” should be replaced by “the percentage”
Done
Reviewer 2 Report
The paper proposed by Perez-Garcia and colleagues aimed to investigate the epidemiology of canine leptospirosis in dogs in Colombia.
The paper is well written and the research is well conducted.
I don't have any comments about the research, but some references need to be added in the introduction and in the discussion.
Missing references:
-Miller, M.D.; Annis, K.M.; Lappin, M.R.; Lunn, K.F. Variability in Results of the Microscopic Agglutination Test in Dogs with Clinical Leptospirosis and Dogs Vaccinated against Leptospirosis. J. Vet. Intern. Med. 2011, 25, 426–432.
-Scanziani, E.; Origgi, F.; Giusti, A.M.; Iacchia, G.; Vasino, A.; Pirovano, G.; Scarpa, P.; Tagliabue, S. Serological survey of leptospiral infection in kennelled dogs in Italy. J. Small Anim. Pract. 2002, 43, 154–157.
-Cilia, G.; Fratini, F.; Turchi, B.; Ebani, V.V.; Turini, L.; Bilei, S.; Bossù, T.; De Marchis, M.L.; Cerri, D.; Bertelloni, F. Presence and Characterization of Zoonotic Bacterial Pathogens in Wild Boar Hunting Dogs (Canis lupus familiaris) in Tuscany (Italy). Animals 2021, 11, 1139. https://doi.org/10.3390/ani11041139
- Ayral, F.C.; Bicout, D.J.; Pereira, H.; Artois, M.; Kodjo, A. Distribution of Leptospira serogroups in cattle herds and dogs in France. Am. J. Trop. Med. Hyg. 2014, 91, 756–759.
- Renaud, C.; Andrews, S.; Djelouadji, Z.; Lecheval, S.; Corrao-Revol, N.; Buff, S.; Demont, P.; Kodjo, A. Prevalence of the Leptospira serovars bratislava, grippotyphosa, mozdok and pomona in French dogs. Vet. J. 2013, 196, 126–127.
Moreover, the Reference section must be revised following Journal guidelines
Author Response
REVIEWER 2
Comments and Suggestions for Authors
The paper proposed by Perez-Garcia and colleagues aimed to investigate the epidemiology of canine leptospirosis in dogs in Colombia. The paper is well written, and the research is well conducted. I don't have any comments about the research, but some references need to be added in the introduction and in the discussion.
Missing references:
-Miller, M.D.; Annis, K.M.; Lappin, M.R.; Lunn, K.F. Variability in Results of the Microscopic Agglutination Test in Dogs with Clinical Leptospirosis and Dogs Vaccinated against Leptospirosis. J. Vet. Intern. Med. 2011, 25, 426–432.
-Scanziani, E.; Origgi, F.; Giusti, A.M.; Iacchia, G.; Vasino, A.; Pirovano, G.; Scarpa, P.; Tagliabue, S. Serological survey of leptospiral infection in kennelled dogs in Italy. J. Small Anim. Pract. 2002, 43, 154–157.
- Ayral, F.C.; Bicout, D.J.; Pereira, H.; Artois, M.; Kodjo, A. Distribution of Leptospira serogroups in cattle herds and dogs in France. Am. J. Trop. Med. Hyg. 2014, 91, 756–759.
These references are now present in the text. References are cited using the ACS style
Round 2
Reviewer 1 Report
Minor
Fig.1: I kindly ask the authors to check the number of total dogs per region, because the data reported in fig.1 does not match the data shown in table 2. For example: for Uraba the total number is not 121 but 156 (17+15+11+16+27+30+8+32). The same for the other regions.
Line 147-148 and line 268: Please modify the sentence based on the new corrected data in Table 3.
Author Response
Thank you for your comments. We have included them in the text of the manuscript with red ink.
Reviewer 2 Report
Authors replied to all my comments
Author Response
There were no comments in this rounds from reviewer 2.